# How much is too much? A medication use evaluation of VA ICU sedation practice during the COVID-19 pandemic

Ian C. Murphy[1], Kelly Bryan[2], Muriel Burk[2], Rong Jiang[2], Francesca Cunningham[2], Sarah Providence[3], Elizabeth Rightnour[4], Sarah Zavala[5], Kathleen Morneau[6], Trisha Exline[4], Stacey Rice[7], Travis Schmitt[8], Kelly Drumright[9], Jennifer Lee[10], BreAnna Davids[11], Tram Guilbeault[12], Brooke Klenosky[13], Ann-Marie Sutherland[9], Abbie Rosen[14], Lauren Ratliff[8], Kenneth Bukowski[2], Margaret A. Pisani[1], Andrew Franck[14], Mark Wong[6], Preston Witcher[7], Kathleen M. Akgün[1,15]*

1 Yale University School of Medicine, New Haven, Connecticut, United States of America, 2 Veterans Affairs Center for Medication Safety-Pharmacy Benefits Management Services; Washington, District of Columbia, United States of America, 3 Veterans Affairs Pittsburgh Healthcare System, Pittsburgh, Pennsylvania, United States of America, 4 Newton D. Baker VA Medical Center, Martinsburg, West Virginia, United States of America, 5 Jesse Brown Veterans Affairs Medical Center, Chicago, Illinois, United States of America, 6 South Texas Veterans Health Care System, San Antonio, Texas, United States of America, 7 United States of America Department of Veterans Affairs, Eastern Colorado Health Care System, Aurora, Colorado, United States of America, 8 Veterans Healthcare System of the Ozarks, Fayetteville, Arkansas, United States of America, 9 Veterans Affairs Tennessee Valley Healthcare System, Nashville, Tennessee, United States of America, 10 Veterans Affairs Long Beach Health Care System, Long Beach, California, United States of America, 11 Ralph H. Johnson Veterans Affairs Medical Center, Charleston, South Carolina, United States of America, 12 Augusta Veterans Affairs Medical Center, Augusta, Georgia, United States of America, 13 Corporal Michael J. Crescenz Veterans Affairs Medical Center, Philadelphia, Pennsylvania, United States of America, 14 Malcom Randall Veterans Affairs Medical Center, North Florida/South Georgia Veterans Health System, Gainesville, Florida, United States of America, 15 Veterans Affairs Connecticut Healthcare System, West Haven, Connecticut, United States of America

* kathleen.akgun@yale.edu

## Abstract

### OBJECTIVES

Early data suggested higher sedative requirements for ventilated COVID+ patients, deviating from established guidelines. We assessed the relationship between sedative use and outcomes in mechanically ventilated Veterans during the COVID-19 pandemic.

### Design

Retrospective Medication Use Evaluation

### Setting

National Sample of 13 Distinct VA Medical Center Intensive Care Units

**Data availability statement:** Due to U.S. Department of Veterans Affairs (VA) regulations and ethics agreements, the analytic data sets used for this study are not permitted to leave the VA firewall without a Data Use Agreement. This limitation is consistent with other studies based on VA data. However, VA data are made freely available to researchers with an approved VA study protocol. For more information, please visit https://www.virec.research.va.gov or contact the VA Information Resource Center at VIReC@va.gov.

**Funding:** The author(s) received no specific funding for this work.

**Competing interests:** NO authors have competing interests.

## Patients

Critically ill Veteran patients requiring mechanically ventilation for ≥2 days

## Interventions

None.

## Measurements and main results

The proportion of patients receiving fentanyl, midazolam and propofol was higher during COVID years. Compared with pre-COVID, median fentanyl dose was higher during Years 1 and 2 (1575mcg [(IQR) 1000–1650] vs. 1900 [1250–3000] vs. 1910 [1150–3500]). Adjuvant antipsychotics use was relatively low but tended to increase over time (pre = 10.5% vs. Year 1 = 12.3% vs. Year 2 = 14.1%). Most patients started on antipsychotics in the ICU were continued on the drug after extubation. Mortality was higher during COVID years (pre = 26.9% vs. 1 = 36.8% and 2 = 35.9%). In stratified analyses by COVID status years 1–2 (n = 79, 27%), a higher proportion of COVID+ patients received fentanyl (96% vs. 84%) and propofol (90% vs. 77%) and at higher doses (fentanyl = 1650mcg vs. 2688mcg median cumulative dose; propofol maximum infusion rate = 30 mc/kg/min (20–50) vs. 40 (25–50)). Sedative doses were similar to pre-COVID among non-COVID patients. Anti-psychotics were more frequently continued post extubation among COVID+ (34.6% vs. non-COVID+ = 14.9%). COVID+ patients were also less likely to have awakening and breathing trials at 48 hours after intubation (18% vs. 46%).

## Conclusions

Sedative use and dosing increased during the first two years of COVID compared to pre-COVID, especially for COVID+ patients. The sustained elevated levels of fentanyl use in Year 2 suggests possible 'therapeutic creep' away from guideline-concordant practices for COVID+ patients. Antipsychotic prescription during intubation and following extubation was also more common among COVID+. These findings could inform development and implementation of safer sedation practices across VA ICUs during respiratory pandemics.

## Introduction

The management of critically ill patients on mechanical ventilation in the intensive care unit (ICU) presents a unique challenge for providers to find a balance between analgesia/anesthesia for ventilator synchrony and wakefulness for early, successful ventilator liberation. Excessive sedative exposure, particularly use of benzodiazepines has been shown to increase time on ventilator, ventilator-associated pneumonia (VAP), delirium risk, ICU, hospital lengths of stay, and patient mortality [1, 2]. Patients who received high doses of sedatives are also at risk for new prescription

of antipsychotics while in the ICU [3]. Initiation of these medications is frequently continued beyond ICU stay and often inappropriately at time of discharge [4].

Recognizing the profound impact of sedative choices and practices on patient outcomes, the Pain, Agitation, Delirium, Immobility and Sleep Disruption (PADIS) guidelines have emerged as a pivotal framework for optimizing care and enhancing safety [5]. Addressing these elements comprehensively is imperative, as they are interconnected and contribute synergistically to optimize patient outcomes. Indeed, adherence to PADIS principles is associated with improved patient outcomes by reducing complications such as VAP, length of ICU and hospital stays, and mortality rates [6].

The unprecedented global outbreak for the novel coronavirus (COVID-19) pandemic created a significant challenge to consistent guideline adherence. As the pandemic unfolded, increasing demands were placed on ICUs to manage greater numbers of patients with acute respiratory distress syndrome (ARDS) and mechanically ventilated patients compared with the experiences preceding the COVID-19 pandemic. Early data suggested that ventilated, COVID-infected patients (COVID+) with ARDS required higher than expected doses of sedative medications due to their lung pathology, potentially exceeding current practice standards [7].

While guideline revisions evolve, the broader implications of sedative choices for COVID+ remain unknown. In response, we developed a multi-site Medication Use Evaluation (MUE) quality improvement project to describe the relationship between sedative use and outcomes for mechanically ventilated Veterans. We explored three distinct time points to assess how established sedative and analgesic guidelines were followed.

## Materials and methods

### Design

A VA-based multi-site MUE was conducted across 13 geographically diverse VA Medical Centers with local champions. Data were accessed and abstracted through retrospective chart review from 01/04/2023–31/08/2023. Patient identifiers were maintained securely within VA systems for review purposes, and all data used for analysis were de-identified or aggregated in accordance with VA policies. We examined data across a 3-year period (March 2019-March 2022), three distinct eras: pre-COVID (March 2019-February 2020), COVID Year 1 (March 2020-February 2021), COVID Year 2 (March 2021-March 2022). This collaborative effort involved clinical pharmacists, intensivists, statisticians and the VA Pharmacy Benefits Management Services Center for Medication Safety (VA MedSAFE).

### Ethical approval and informed consent

The project was reviewed and approved by the Edward Hines VA Hospital IRB Committee as a non-research QI/QA protocol and determined to be a non-research QI/QA protocol.

### MUE target population

All mechanically ventilated patients during each time frame were screened via electronic health record (EHR) review. We included VA ICU patients who received care for ≥1 calendar day and who required mechanical ventilation for ≥2 calendar days. Exclusion criteria were patients who: died or were transferred within 48 hours of admission, were transferred into the ICU from a non-VA facility, were physically located in the ICU but did not receive ICU care, were ventilated for >14 days or those with a pre-existing tracheostomy or pre-existing chronic ventilator use.

### Data sources and collection

At each participating site, data were retrospectively abstracted from chart review conducted between 01/04/2023 and 31/08/2023 and entered into a secure, encrypted InfoPath server hosted within VA systems. A pilot abstraction tool was developed and iteratively refined to identify clinical variables from the EHR. Team members participated in pilot

development and were trained on data collection and entry procedures. During review and data entry, site-specific patient lists containing identifiers were maintained within the VA protected environment and accessible only to authorized personnel with proper permissions. Each site reviewed approximately 150–200 patient records. Prior to final analysis, data were screened for irregular or implausible values.

## Primary outcomes

The primary outcome was median dose of sedative and opioid exposure for patients. Medication use was divided into commonly used opioids (fentanyl, hydromorphone, methadone, morphine, oxycodone) and sedatives (dexmedetomidine, diazepam, ketamine, lorazepam, midazolam, and propofol). Opioids were converted to morphine milligram equivalents (MME) doses [8, 9]. Infusion rates were manually collected from clinical information systems used at participating sites and validated through chart review and bar code medication administration (BCMA) data when available. The highest and lowest daily infusion rates were collected per day and extrapolated as a maximum potential exposure and minimum potential exposure per 24 hours of drug infusion. Highest and lowest daily infusion rates were collected per day for each drug. While data on all drugs were collected for the first 14 ICU days, we only presented median rates within the first 48 hours of mechanical ventilation for patients with rate data abstracted due to variability and missingness with longer observation time (Fig 1).

Use of antipsychotic agents was assessed during ICU stay, overall hospitalization and after discharge. Antipsychotic agents of interest included haloperidol, quetiapine, olanzapine, and aripiprazole. Exposure to antipsychotics was evaluated daily while the patient was intubated and up to three days after extubation, continuation at time out of ICU transfer, at hospital discharge, and at 3 months after discharge.

## Secondary Outcomes

Secondary outcomes included mortality, delirium prevalence, new antipsychotic prescription and ileus, use of reversal agents [10, 11]. Sedation depth was measured using daily Richmond Agitation Sedation Score (RASS) for highest and

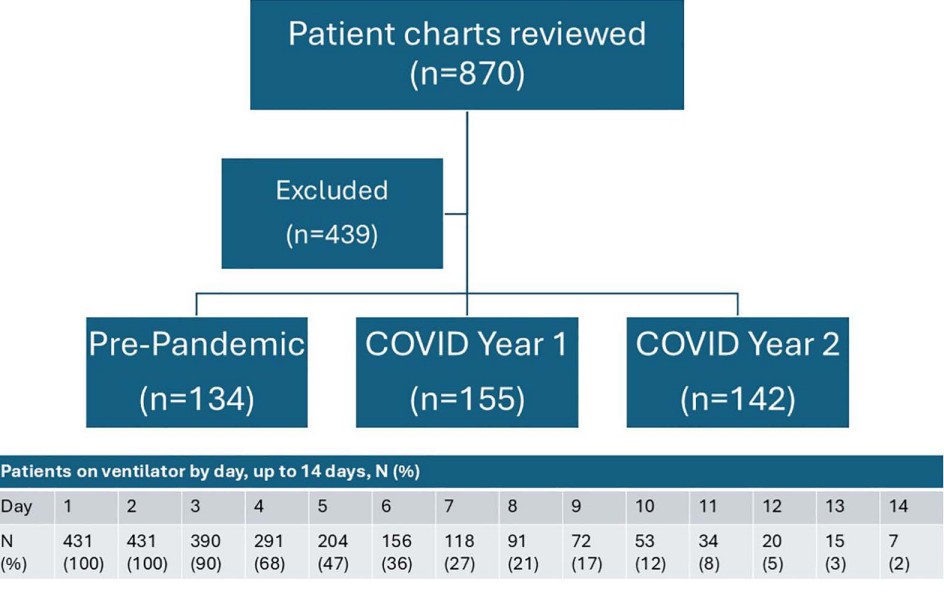

**Fig 1. Breakdown of Patient Inclusion/ Exclusion and time on ventilator.**

lowest RASS score/day. Patient data was grouped into categories for daily RASS scoring if at least one reading fell into pre-defined categories (deep = −2 to −5; goal = −1or 0; agitated=+1 to +4). This categorical separation was based on studies evaluating percentage of time at a goal RASS score [12]. Fluctuations were defined if at least two of the RASS categories were identified on the same day. These categories were selected due to known risks of large fluctuations in RASS score during a 24-hour period [13]. We assessed rates of spontaneous breathing trial (SBT) documentation at 24 and 48 hours after intubation. Additional data was collected on rates of spontaneous awakening trials (SAT) or evidence of sedative dose reduction at 24 and 48 hours.

Delirium assessment was gathered via documentation of Confusion Assessment Method (CAM-ICU). [14, 15] Patients were evaluated for a documented positive CAM-ICU on each ICU day or if the information was not available. Delirium ascertainment continued up until three days after extubation. Patient comfort was assessed via daily collection of Critical Pain Observation Tool (CPOT) [16] or Numeric Rating Scale (NRS) [17]. The highest and lowest values per day were collected for entirety of ICU stay (up to 14 days) and up to three days after extubation.

Patients were followed until death or up until three days after ICU transfer as data was available. Additional analysis was performed to assess differences between COVID+ and non-COVID+ patients during Years 1 and 2 in and primary and secondary outcomes.

## Covariates

Baseline data included ICU admission age, gender, weight (kg) at time of admission, race, ethnicity, comorbid medical conditions (using Charlson comorbidity index (CCI) score) [18, 19] primary location of care (MICU, CCU, SICU, or other location) and COVID+ status and COVID- ARDS. Covariates for processes of care included ventilator settings, vasopressor use, SAT, SBT, need for reintubation, use of restraints, and post-ICU disposition, use of specific reversal agents (methylnaltrexone, flumazenil, and sugammadex use).

## Statistical analysis

Descriptive statistics were used to assess baseline patient characteristics. Data on a continuous scale were presented as median (± interquartile range) while data on a categorical scale was presented as frequencies or percentages. We chose the median value for sedative infusion rates due to available data in CPRS and clinical information systems which include documentation of rate, which is not normally distributed, rather than total dose.

Based on available information [20, 21], we estimated that there would be a 30% increase in the median amount of sedative drug received by all patients during the COVID-19 pandemic. To achieve appropriate power to detect this clinically meaningful difference, we estimated that 200 patients should be included in each time period. Following data eligibility screening, we met 72% of our expected sample size.

Secondary exploratory analyses compared sedative and opioid exposure by COVID+ status although statistical testing for differences by patient groups was beyond the scope of the MUE quality improvement structure.

## Results

Among 870 patients across the three time periods, 431 (49.5%) with complete data during the first 48 hours were included for chart review; pre-COVID n = 134, Year 1 n = 155, Year 2 = 142 (Fig 1). Demographic characteristics were not significantly different across the time periods (Table 1) with median age = 69, 71, 71, proportion male = 96%, 97%, 95% and identified as white race = 69%, 71%, 70%, respectively. Median CCI was highest pre-COVID = 5.3 vs. Year 1 = 4.7 and Year 2 = 4.0. The medical ICU was the primary service for most patients (78%, 83%, 70% respectively). Mortality was higher during COVID periods vs. pre-COVID (27% vs. Year 1 = 37%, Year 2 = 36%).

Fentanyl was the most frequently used opioid; median 48-hour dose increased from pre to COVID years (Table 2). Pre-COVID patients (n = 82/109 (75%) with rate data) received a median fentanyl dose of 1575mcg (1000, 2650) vs.

**Table 1. Demographic Information for Patient Cohorts by COVID year.**

| Variable | Pre COVID (n = 134) | Year 1 (n = 155) | Year 2 (n = 142) |
|---|---|---|---|
| Age Median (IQR) | 69 (62, 74) | 71 (63,75) | 71 (64,76) |
| Male Sex, N (%) | 129 (96.2) | 150 (96.7) | 135 (95.0) |
| Race, N (%) | | | |
| Am. Indian | 0 (0) | 0 (0) | 1 (0.7) |
| Asian | 0 | 0 | 0 |
| Black | 34 (25.3) | 43 (27.7) | 34 (24) |
| Hispanic | 4 (2.9) | 2 (1.2) | 4 (2.8) |
| White | 92 (68.6) | 110 (71) | 100 (70.4) |
| Charlson Comorbidities Median (IQR) | 5.3 (2, 7) | 4.7 (2, 6) | 4.0 (2, 5) |
| COVID+, n (%) | 0 (0) | 42 (27.1) | 37 (26.1) |
| Care in MICU, n (%) | 104 (77.6) | 128 (82.5) | 100 (70.4) |
| Discharge Disposition, n (%) | | | |
| Skilled Nursing Facility | 1 (0.7) | 2 (1.2) | 1 (0.7) |
| Inpatient Rehabilitation | 3 (2.2) | – | 4 (2.8) |
| Step-down ICU | 33 (24.6) | 32 (20.6) | 26 (18.3) |
| Traditional medical floor unit | 52 (38.8) | 56 (36.1) | 49 (34.5) |
| Different ICU (SICU/CCU) | 1 (0.7) | – | – |
| Home | 6 (4.4) | 6 (3.8) | 11 (7.7) |
| Mortality n (%) | 36 (26.9) | 57 (36.8) | 53 (35.9) |

**Table 2. Sedative and Fentanyl Exposure in Mechanically Ventilated Veterans during the first 48 Hours after Intubation.**

| Parameter | Pre COVID (n = 134) | Year 1 Overall (n = 155) | Year 2 Overall (n = 142) |
|---|---|---|---|
| Fentanyl, n (%) | 109 (81) | 132 (85) | 125 (88) |
| Fentanyl, Median Total Dose, mcg (IQR)* | 1575 (1000, 2650) | 1900 (1250, 3000) | 1910 (1150, 3500) |
| Midazolam, n (%) | 36 (27) | 46 (30) | 46 (32) |
| Midazolam, Median Max rate, mg/hr (IQR)* | 4 (2.0, 4.0) | 4 (2.5, 9.0) | 3.5 (2.0, 6.0) |
| Dexmedetomidine, n (%) | 67 (50) | 66 (43) | 61 (43) |
| Dexmedetomidine, Max rate, mcg/kg/hr (IQR)* | 0.7 (0.4, 1.0) | 0.65 (0.4, 0.9) | 0.6 (0.5, 1.0) |
| Propofol, n (%) | 99 (74) | 121 (78) | 118 (83) |
| Propofol, Med Max rate, mcg/kg/min (IQR)* | 33 (20, 50) | 30 (20,45) | 35 (20, 50) |
| Neuromuscular blocker, n (%) | 5 (3.7) | 27 (17.4) | 25 (17.6) |
| **Processes of ventilator care on intubation day 2** | | | |
| Spontaneous Awakening Trial or Sedative Dose Reduction, n (%) | 69 (53.4) | 55 (36.8) | 56 (40.1) |
| Spontaneous Breathing Trial, n (%) | 33 (25.5) | 33 (22.1) | 34 (24.4) |
| Pts w/ Ileus, n (%) | 7 (5.2) | 11 (7.1) | 9 (6.3) |

*Rate data available for subset of patients receiving sedative medication, representing:

Pre COVID: fentanyl=82/109 (75%), midazolam=14/36 (39%), dexmedetomidine=40/67 (60%), propofol=83/99 (84%)).

Year 1: fentanyl=108/132 (82%), midazolam=28/46 (61%), dexmedetomidine=34/66 (52%), propofol=102/121 (84%)).

Year 2: fentanyl=99/125 (79%), midazolam=23/46 (50%), dexmedetomidine=31/61 (51%), propofol=103/118 (87%)).

Year 1 = 1900 (1250, 3000) and Year 2 = 1910 (1150, 3500); doses varied with COVID+ status during years 1 and 2: COVID+=2605–3000 vs. non-COVID = 1575–1675 (Table 3). Median maximum propofol infusion rate at 24 and 48 hours was similar across the three time periods, but the percentage of patients still receiving propofol after 48 hours increased over time (62%, 66%, 73%, respectively). A similar trend was seen in median maximum midazolam infusion rate with no difference among groups by year (Table 2) but higher midazolam by COVID+ status. Additionally, a significantly higher portion of patients were treated with a midazolam infusion after 48 hours (10%, 18%, 16%, respectively). In contrast, dexmedetomidine use decreased from pre- vs. COVID years and appeared to be used differently in Year 1 vs. Year 2 by COVID status. There was minimal use of reversal agents across time periods. The percentage of patients diagnosed with ileus was also low and similar across time periods, 5.2% in pre-COVID, 7.1% in Year 1 and 6.3% in Year 2.

Median number of ventilator days with ≥1 RASS assessment was 4 (3,7). Pre-COVID patients had the highest percentage of days with a RASS score at goal (79.8%); COVID Year 2 patients also had 78.8% of days with goal RASS. In contrast, Year 1 had a lower percentage of patients with a goal RASS (66.4%). Across all time periods a total of 33.2% of patients had at least one day with a positive CAM-ICU score. Similar to RASS fluctuation data, there was an increased number of median CAM-ICU+ days for the COVID Year 1 cohort=5 days (4,8), compared to pre-COVID and COVID Year 2 which each had a median of 4 days.

Differences in rates of documented SAT and SBT data also emerged in comparison by time period. A higher percentage of pre-COVID patients had a documented SAT or reduction in sedative dose on Day 2 of intubation (53.4%) compared to Year 1 (36.8%) and Year 2 (40.1%). Rates of SBT at Day 2 of intubation were more consistent across the 3 cohorts (25.5%, 22.1%, and 24.4%, respectively).

Within the entire cohort, a total of 12.3% of patients received an antipsychotic medication, most commonly quetiapine (43%) and haloperidol (30%). Although antipsychotic use was relatively low, it tended to increase from pre-COVID to Year 2 (pre = 10% vs. Year 1 = 12% vs. Year 2 = 14%) (Table 4). Most patients started on antipsychotics in the ICU were continued on the drug at time of transfer out of the ICU, 12/14 (85%) patients pre-COVID, 17/19 (89%) in Year 1 and 20/20 (100%) in Year 2, and 20–30% were continued at hospital discharge in each time period. By 3 months post discharge, all newly prescribed antipsychotics were discontinued.

**Table 3. COVID vs. Non-COVID Group Stratification during Years 1 and 2.**

| | Overall | | Year 1 | | Year 2 | |
|---|---|---|---|---|---|---|
| | COVID+ (n = 79) | Non COVID (n = 218) | COVID+ (n = 42) | Non COVID (n = 113) | COVID+ (n = 37) | Non COVID (n = 105) |
| Fentanyl | 74 (96) | 183 (84) | 38 (90) | 94 (83) | 36 (97) | 89 (85) |
| Fentanyl, Median Total Dose, mcg (IQR)* | 2688 (1722.5, 4025) | 1650 (1080, 2800) | 2605 (1840, 3820) | 1675 (1080, 3000) | 3000 (1658, 4700) | 1575 (1050, 2575) |
| Midazolam, n (%) | 30 (38) | 62 (28) | 13 (31) | 33 (29) | 17 (46) | 29 (28) |
| Dexmedetomidine, n (%) | 34 (43) | 93 (43) | 21 (50) | 45 (40) | 13 (35) | 48 (46) |
| Propofol, n (%) | 71 (90) | 168 (77) | 35 (83) | 86 (76) | 36 (97) | 82 (78) |
| Propofol, Med Max rate, mcg/kg/min (IQR)* | 40 (25-50) | 30 (20-50) | 37 (18, 50) | 30 (20, 45) | 45 (30, 50) | 30 (20, 50) |
| Neuromuscular blocker, n (%) | 38 (48) | 14 (6) | 16 (38) | 11 (10) | 22 (59) | 3 (3) |
| **Processes of ventilator care on intubation day 2 and outcomes** | | | | | | |
| SAT or Sedative Dose Reduction, n (%) | 14 (18) | 97 (46) | 7 (5) | 48 (33) | 7 (5) | 49 (35) |
| Spontaneous Breathing Trial, n (%) | 7 (9) | 60 (28) | 5 (3) | 28 (19) | 2 (1) | 32 (23) |
| Ileus, n (%) | 3 (4) | 17 (8) | 1 (2) | 10 (9) | 2 (5) | 7 (6) |

IQR=interquartile range; SAT=Spontaneous awakening trial

Year 1: fentanyl=108/132 (82%), midazolam=28/46 (61%), dexmedetomidine=34/66 (52%), propofol=102/121 (84%)).

Year 2: fentanyl=99/125 (79%), midazolam=23/46 (50%), dexmedetomidine=31/61 (51%), propofol=103/118 (87%)).

**Table 4. Delirium and Antipsychotic Use During and Post-ICU Discharge for Mechanically Ventilated Patients Across 3 Time Periods.**

| Parameter | Pre COVID (n = 134) | Year 1 (n = 155) | Year 2 (n = 142) |
|---|---|---|---|
| Intubated CAM-ICU+ Days Median (IQR) | 4 (3, 7) | 5 (4, 8) | 4 (3, 7) |
| Extubated CAM-ICU+ Days Median (IQR) | 1 (1, 2) | 2 (1, 3) | 1 (1, 3) |
| Prescribed an Antipsychotic in the ICU, n (%) | 14 (10.5) | 19 (12.3) | 20 (14.1) |
| Antipsychotic continued post-ICU, n (%) | 12 (9.0) | 17 (11.0) | 20 (14.1) |
| Antipsychotic continued at discharge n (%) | 4 (3.0) | 5 (3.2) | 3 (2.1) |

When stratified by COVID+, COVID+ patients' median age in years was 72 vs. 70 and median CCI was lower (3.6) vs. non-COVID+ (4.6); ARDS was identified in 78% of COVID+ compared with 4% of uninfected. Secondary analysis demonstrated higher drug exposure for COVID+ (Table 3). During COVID years and compared with non-COVID patients, a higher proportion of COVID+ received fentanyl (84% vs. 96%) and propofol (77% vs. 90%) at any time during the first 14 ICU days (Table 3). COVID+ also received higher doses of fentanyl (median dose = 1650mcg vs. 2688 mcg), midazolam (Year 1 median dose COVID+=7 [interquartile range 3, 10] vs. 4 [2, 8]; Year 2 COVID+=5 [2, 6] vs. 3 [3, 4]) and propofol (median max 48-hr infusion rate = 30 mcg/kg/min (20–50) vs. 40 mcg/kg/min (25–50)) during the same time period. Dexmedetomidine doses were similar between the two groups (Year 1 median dose COVID+=0.6 [0.4, 0.8] vs. 0.7 [0.3, 1.0]; Year 2 COVID+=0.5 [0.5, 0.5] vs. 0.7 [0.5, 1.0]). Anti-psychotics were more frequently continued among COVID+ patients post extubation vs. non-COVID (34.6% vs. 14.9%). In contrast to the higher rates of sedation, COVID+ patients had lower rates of SATs and SBTs at 48 hours post intubation. Only 18% of COVID+ patients had a documented SAT or sedative dose reduction compared to 45.8% of non-COVID patients. A similar trend was found in SBT percentages with 9.2% of COVID+ patients having documentation of a breathing trial compared to 28.2% of non-COVID patients. Mortality was higher in COVID+ patients, 71% compared to non-COVID patients 24%, across the time periods of interest.

## Discussion

This MUE demonstrates shifts in sedation practices within the VA healthcare system during the initial two years of the COVID-19 pandemic, indicating an increase in sedation use and dosing compared to the pre-COVID era. When stratified by COVID status, the increased sedative use was for patients with COVID compared with the non-COVID patients. Anecdotally, higher sedation was likely needed to support patients on the ventilator although we were unable to report ARDS severity in this work. Staffing ratios appeared to be relatively stable, although this was not specifically measured. Our data also illustrate that these shifts were occurring across a nationally representative sample of patients in the VA.

Our findings align with observations from smaller, non-VA studies suggesting a general trend towards increasing sedative use due to the unique challenges posed by COVID-19 in critical care settings [7, 20, 22]. Similar to one single-center study including 86 COVID+, more than 50% of COVID+ received sedative doses that exceeded the upper limit of hospital guidelines [21]. Evaluating consequences of deviations from sedation guidelines is ongoing. A large retrospective cohort study demonstrated an association between higher doses of sedatives given to COVID+ patients and increased mortality [20].

Multiple potential reasons for sedation escalation have been proposed. The need to achieve ventilator synchrony in the setting of exaggerated inflammatory response and complex respiratory mechanics from COVID+ ARDS likely contributed to higher sedation [23]. Moreover, COVID+ ARDS compared to age-matched controls required higher doses of sedatives, oftentimes more benzodiazepines to achieve the same level of sedation [24]. Factors more extrinsic to the COVID-19 pandemic like staffing pressures, lack of adequate personal protective equipment (PPE), medication shortages, provider fatigue and incomplete understanding of the disease process also could have significantly impacted sedation practices [25].

The escalated use of sedatives, particularly midazolam, underscores the deviation from guideline-driven management strategies for ventilated COVID+ patients. While benzodiazepines were originally part of first line sedation, growing literature has demonstrated increased patient harms with use, and thus advocating for minimal midazolam exposure for sedated patients [26]. While these changes may reflect a potential adaptation to the heightened requirements for mechanical ventilation, they potentially contributed to poorer patient outcomes. Recent studies demonstrated increased mortality for COVID+ patients in hospitals that used primarily opioids and benzodiazepines compared to those using propofol and opioids for sedation [27, 28]. We noted an increased percentage of patients exposed to midazolam in COVID Years 1 and 2 compared to pre-COVID, although this was more frequently occurring for patients with COVID and appeared to be returning to pre-COVID practices for patients without COVID by Year 2. We expected to observe higher mortality during COVID Year 1, compared to pre-COVID given the overwhelming nature of initial COVID surges. However, we noted unexpected sustained increased mortality in Year 2, despite a lower median CCI score and an assumption that providers had become more adept at managing COVID ARDS. While multiple explanations of this sustained worsened mortality exist, we speculate that the ongoing increased sedative, particularly benzodiazepine use, may have played a role in negatively affecting patient outcomes. We propose that returning to sedation stewardship principles could improve ICU care for patients on the ventilator.

Our data also indicate a trend toward disparate use of antipsychotic medications during the first two years of the COVID-19 pandemic. During all time periods, antipsychotics started in the ICU were rarely discontinued at time of transfer. Of note, antipsychotic prescription during intubation and following extubation were more commonly seen among COVID+ patients. The pandemic's onset marked a noticeable increase in the incidence of delirium among ICU patients, particularly those exposed to benzodiazepines [29]. Moreover, COVID+ correlated with a reduced rate of spontaneous awakening and breathing trials, critical components in the pathway towards extubation, sedation liberation and ICU discharge. Higher rates of delirium may have led to increased antipsychotic prescription, and lack of clear safeguards prevented appropriate medication reconciliation at time of transfer. This relationship underscores the complex interplay between sedative management, patient condition, and the resultant neuropsychiatric outcomes, highlighting the nuanced challenges faced in the pharmacological care of COVID-19 patients.

The persistent high utilization of fentanyl and midazolam into the second year of the pandemic hints at a phenomenon of 'therapeutic creep,' where there is a gradual and sustained shift in clinical practice away from existing guideline-concordant sedation strategies. This phenomenon is not well described in the literature but is certainly felt within the medical community [30]. In cases of "sedation creep", providers become comfortable with higher doses or duration of medications and continue those practices, even beyond the stressors of a pandemic surge. However, we only characterize sedative use during the first two years of the COVID pandemic and do not have "post-pandemic" practices to evaluate whether sedation creep is occurring. We also are unable to characterize ARDS severity or prone positioning, which would also influence sedation needs. Ongoing efforts are needed to promote guideline-concordant ICU sedation practices for patients on the ventilator.

Several important limitations were identified with this MUE quality improvement project. First, we restricted data analysis to sedation and opioid use during the initial 48 hours of mechanical ventilation; we were unable to get hour-by-hour infusion rates, potentially overlooking changes in medication regimens even during that timeframe that could impact patient outcomes. We also only had complete data for the first 48 hours of all included patients due to the rapid decrease in ventilator use subsequently (Fig 1), limiting our ability to comprehensively evaluate ICU medication use. Additionally, this was a retrospective chart review methodology, limiting our data to what was documented in the EHR. Furthermore, the findings from the MUE are meant to inform Veteran care in the ICU rather than evaluating for statistically significant differences in groups included in this project; our findings are also potentially not generalizable to non-VA settings. Moreover, the rate of exclusion stemming from mechanical ventilation durations less than 2 days or exceeding 14 days may introduce selection bias, potentially skewing the representativeness of the cohort and limiting the extrapolation of results

to patients with prolonged ventilation requirements. Finally, our findings are only descriptive frequencies and do not include patient-centered outcomes to better understand how medication use impacted clinical experiences for patients with ARDS.

## Conclusion

Our MUE findings advocate for sedation guideline realignment for ventilated patients and de-prescribing strategies upon transfer out of the ICU to minimize the unnecessary continuation of neuropsychiatric medications. The insights gained from this comprehensive analysis not only shed light on the evolving sedation practices because of the COVID-19 pandemic but also serve as a valuable foundation for the development and implementation of safer, evidence-based sedation guidelines across VA ICUs. Moving forward, it is imperative to integrate these lessons into clinical protocols to enhance patient outcomes and align practices with the latest clinical evidence and standards in VA ICUs.

## Author contributions

**Conceptualization:** Ian C. Murphy, Kelly Bryan, Muriel Burk, Kathleen M. Akgün.

**Data curation:** Ian C. Murphy, Muriel Burk, Rong Jiang, Sarah Providence, Elizabeth Rightnour, Sarah Zavala, Kathleen Morneau, Trisha Exline, Stacey Rice, Travis Schmitt, Kelly Drumright, Jennifer Lee, BreAnna Davids, Tram Guilbeault, Brooke Klenosky, Ann-Marie Sutherland, Abbie Rosen, Lauren Ratliff, Andrew Franck, Mark Wong, Preston Witcher, Kathleen M. Akgün.

**Formal analysis:** Kelly Bryan, Rong Jiang.

**Funding acquisition:** Muriel Burk, Francesca Cunningham.

**Investigation:** Ian C. Murphy, Kelly Bryan, Francesca Cunningham, Sarah Providence.

**Methodology:** Ian C. Murphy, Kelly Bryan, Muriel Burk, Francesca Cunningham, Kathleen M. Akgün.

**Project administration:** Muriel Burk, Francesca Cunningham, Sarah Providence, Kenneth Bukowski.

**Supervision:** Muriel Burk, Kathleen M. Akgün.

**Writing – original draft:** Ian C. Murphy, Kelly Bryan, Muriel Burk, Rong Jiang, Francesca Cunningham, Sarah Providence, Elizabeth Rightnour, Sarah Zavala, Kathleen Morneau, Trisha Exline, Stacey Rice, Travis Schmitt, Kelly Drumright, Jennifer Lee, BreAnna Davids, Tram Guilbeault, Brooke Klenosky, Ann-Marie Sutherland, Abbie Rosen, Lauren Ratliff, Margaret A. Pisani, Andrew Franck, Mark Wong, Preston Witcher, Kathleen M. Akgün.

**Writing – review & editing:** Kelly Bryan, Muriel Burk, Rong Jiang, Margaret A. Pisani, Kathleen M. Akgün.

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
