## [Decision Letter · Decision Letter 0]

28 Nov 2025

PONE-D-25-50605How Much is Too Much? A Medication Use Evaluation of VA ICU Sedation Practice During the COVID-19 Pandemic?

PLOS ONE

Dear Dr. Akgün,

Thank you for submitting your manuscript to PLOS ONE. After careful consideration, we feel that it has merit but does not fully meet PLOS ONE’s publication criteria as it currently stands. Therefore, we invite you to submit a revised version of the manuscript that addresses the points raised during the review process.

**Thank you for sending your manuscript to  PLOS One. Reviewers are two **
**anesthesiologists with extensive experience in COVID ICU, please address their comments. I am looking forward to revised version of  the manuscript.**

We look forward to receiving your revised manuscript.

Kind regards,

Benjamin Benzon, Ph.D., M.D.

Academic Editor

PLOS ONE

Journal Requirements:

b) If there are no restrictions, please upload the minimal anonymized data set necessary to replicate your study findings to a stable, public repository and provide us with the relevant URLs, DOIs, or accession numbers. Please see http://www.bmj.com/content/340/bmj.c181.long for guidelines on how to de-identify and prepare clinical data for publication. For a list of recommended repositories, please see https://journals.plos.org/plosone/s/recommended-repositories. You also have the option of uploading the data as Supporting Information files, but we would recommend depositing data directly to a data repository if possible.x

3. Please be informed that funding information should not appear in the Acknowledgments section or other areas of your manuscript. We will only publish funding information present in the Funding Statement section of the online submission form. Please remove any funding-related text from the manuscript.

Reviewers' comments:

Reviewer's Responses to Questions

**Comments to the Author**

1. Is the manuscript technically sound, and do the data support the conclusions?

Reviewer #1: Partly

Reviewer #2: Partly

2. Has the statistical analysis been performed appropriately and rigorously?

Reviewer #1: Yes

Reviewer #2: Yes

3. Have the authors made all data underlying the findings in their manuscript fully available?

Reviewer #1: Yes

Reviewer #2: No

4. Is the manuscript presented in an intelligible fashion and written in standard English?

Reviewer #1: Yes

Reviewer #2: Yes

Reviewer #1: Although study is interesting, there are some problems with methodology and interpretation od data.

This retrospective study limitation is just collecting data for 48 h sedation.

It is not clear for me how this narrow time periods support your conclusion.

This study is not patient centred.

You did not reach estimated sample size.

COVID ARSD was unique lung patology, the worst cases that would dye spontaneously breading was put on mechanical ventilation,

and stiff lungs were hard to ventilate.

It was necessary to give more sedation than non COVID ICU patients to achieve smooth ventilation, maybe muscle relaxation was under dosed and sedation overdosed, but still it is not clear to me was you can infer for just 2 days of ICU sedation.

In the first 2 days it is hard to have prognosis (survival or death).

I think conclusion is not supported by data and study design.

Reviewer #2: I would like to see your opinions in discussion section what were the possible reasons for analgosedation increase in Covid patients? Was it too many difficult patients and too little doctors and nurses so they could not take care as usual?

Did you calculate Horowitz index for ARDS severity? What was vaccination status in year 2? Did you put patients in prone position for better ventilation?

All above are possible factors for analgosedation increase.

**Do you want your identity to be public for this peer review?** For information about this choice, including consent withdrawal, please see our Privacy Policy

Reviewer #1: No

Reviewer #2: No

---

## [Author Response · Author response to Decision Letter 1]

9 Dec 2025

please see attached response letter for improved formatting

Reviewer: 1

Comments to the Author

Although study is interesting, there are some problems with methodology and interpretation od data.

Reviewer comment 1:

This retrospective study limitation is just collecting data for 48 h sedation.

Reply 1: We agree that this is a limitation. However, complete data was only available during the first 48 hours; by day 5, more than half of the patients were off of the ventilator. We included these details in figure 1 and in the text, and as additional limitations in the manuscript.

Results, page 10, lines 294-295: “431 (49.5%) with complete data during the first 48 hours were included for chart review”

Discussion, page 16, lines 408-410: “We also only had complete data for the first 48 hours of all included patients due to the rapid decrease in ventilator use subsequently (Figure 1), limiting our ability to comprehensively evaluate ICU medication use.”

Additional table added to figure 1:

Patients on ventilator by day, up to 14 days, N (%)

Day 1 2 3 4 5 6 7 8 9 10 11 12 13 14

N

(%) 431 (100) 431 (100) 390

(90) 291

(68) 204

(47) 156

(36) 118

(27) 91

(21) 72

(17) 53

(12) 34

(8) 20

(5) 15

(3) 7

(2)

Reviewer comment 2:

It is not clear for me how this narrow time periods support your conclusion.

Reply 2: We merely intended to report observations of medication prescribing in a national sample. We feel our conclusions are tempered and seeking to remind providers of the importance of sedation protocols and sedation stewardship.

Reviewer comment 3:

This study is not patient centred.

Reply 3: As a chart review study, we agree we are limited in patient-centered outcomes. We included this as another limitation in the manuscript.

Discussion, page 17, lines 439-441: “Finally, our findings are only descriptive frequencies and do not include patient-centered outcomes to better understand how medication use impacted clinical experiences for patients with ARDS.”

Reviewer comment 4:

You did not reach estimated sample size.

Reply 4: We regret that we did not reach our intended goals for chart review due to exclusion criteria. This was included as a limitation already in the manuscript.

Reviewer comment 5:

COVID ARSD was unique lung patology, the worst cases that would dye spontaneously breading was put on mechanical ventilation, and stiff lungs were hard to ventilate.

It was necessary to give more sedation than non COVID ICU patients to achieve smooth ventilation, maybe muscle relaxation was under dosed and sedation overdosed, but still it is not clear to me was you can infer for just 2 days of ICU sedation.

In the first 2 days it is hard to have prognosis (survival or death).

I think conclusion is not supported by data and study design.

Reply 5: Please see response to comments 1 and 2 above. We also included more context about the stiff lung physiology of COVD ARDS.

Discussion, page 13, lines 372-374: “The need to achieve ventilator synchrony in the setting of exaggerated inflammatory response and complex respiratory mechanics from COVID+ ARDS likely contributed to higher sedation.”

Reviewer: 2

Reviewer comment 1:

I would like to see your opinions in discussion section what were the possible reasons for analgosedation increase in Covid patients? Was it too many difficult patients and too little doctors and nurses so they could not take care as usual?

Reply 1: We speculated about some of these factors that contributed to higher analgosedation for COVID patients, including possible staffing shortages, but more granular assessments of staffing models.

Discussion, page 13, lines 360-363: “Anecdotally, higher sedation was likely needed to support patients on the ventilator although we were unable to report ARDS severity in this work. Staffing ratios appeared to be relatively stable, although this was not specifically measured.”

Reviewer comment 2:

Did you calculate Horowitz index for ARDS severity? What was vaccination status in year 2? Did you put patients in prone position for better ventilation? All above are possible factors for analgosedation increase.

Reply 2: We did not calculate the Horowitz index for ARDS severity. We also were unable to include information on vaccination status or prone positioning. We included these as additional limitations.

Discussion, page 16, lines 421-423: “We also are unable to characterize ARDS severity or prone positioning, which would also influence sedation needs. Ongoing efforts are needed to promote guideline-concordant ICU sedation practices for patients on the ventilator.”

---

## [Editor Report · Decision Letter 1]

18 Dec 2025

How Much is Too Much? A Medication Use Evaluation of VA ICU Sedation Practice During the COVID-19 Pandemic

PONE-D-25-50605R1

Dear Dr. Akgün,

We’re pleased to inform you that your manuscript has been judged scientifically suitable for publication and will be formally accepted for publication once it meets all outstanding technical requirements.

Kind regards,

Benjamin Benzon, Ph.D., M.D.

Academic Editor

PLOS One
---

## [Editor Report · Acceptance letter]

PONE-D-25-50605R1

PLOS One

Dear Dr. Akgün,

I'm pleased to inform you that your manuscript has been deemed suitable for publication in PLOS One. Congratulations! Your manuscript is now being handed over to our production team.

Kind regards,

on behalf of

Dr. Benjamin Benzon

Academic Editor

PLOS One